# LARGE LANGUAGE MODELS AS REALISTIC MICROSERVICE TRACE GENERATORS

## ABSTRACT

Computer system workload traces, which record hardware or software events during application execution, are essential for understanding the behavior of complex systems and managing their processing and memory resources. However, obtaining real-world traces can be challenging due to the significant collection overheads in performance and privacy concerns that arise in proprietary systems. As a result, synthetic trace generation is considered a promising alternative to using traces collected in real-world production deployments. This paper proposes to train a large language model (LLM) to generate synthetic workload traces, specifically microservice call graphs. To capture complex and arbitrary hierarchical structures and implicit constraints in such traces, we fine-tune LLMs to generate each layer recursively, making call graph generation a sequence of easier steps. To further enforce learning constraints in traces and generate uncommon situations, we apply additional instruction tuning steps to align our model with the desired trace features. Our evaluation results show that our model can generate diverse realistic traces under various conditions and outperform existing methods in accuracy and validity. We show that our synthetically generated traces can effectively substitute real-world data in optimizing or tuning systems management tasks. We also show that our model can be adapted to perform key downstream trace-related tasks, specifically, predicting key trace features and infilling missing data given partial traces.

## 1 INTRODUCTION

Computer system workload traces document hardware or software events that occur as applications execute on computing machines, receive requests, process them, and serve responses. Such traces are vital for analyzing complex computer systems and optimizing their CPU, memory, IO and networking resource allocation and management. However, obtaining real-world traces is often hindered by privacy concerns and their general unavailability. As an alternative, synthetic traces provide limitless size and variety, offering significant advantages for testing and analysis, including the ability to simulate challenging conditions like stress-testing environments. While recent advances in generative machine learning, including LSTMs (Sherstinsky, 2020), GANs (Goodfellow et al., 2014), and diffusion models (Ho et al., 2020), have improved synthetic trace generation, these methods typically only generate specific fields, such as the number of requests or resource types (Bergsma et al., 2021), or are confined to fixed-structure traces, like network packets (Jiang et al., 2023; Yin et al., 2022).

In this paper, we show how to use large language models (LLMs), transformer-based (Vaswani et al., 2017) neural networks pre-trained autoregressively on large and diverse text datasets (Brown et al., 2020; Touvron et al., 2023), to generate synthetic workload traces. It has been shown that LLMs can be readily adapted to model a variety of domains besides natural language, such as protein sequences (Shen et al., 2024), code (Roziere et al., 2023), and tabular data (Borisov et al., 2023). LLMs can produce outputs that are well-aligned with user inputs in several flexible ways such as fine-tuning model weights (Ouyang et al., 2022; Wei et al., 2021), and can generalize to new user inputs at inference (Chung et al., 2022; Sanh et al., 2021). Thus, LLMs have the potential to generate synthetic traces that accurately model the structure of real-world traces while following user prompts.

Despite their potential, using LLMs for synthetic trace generation presents significant challenges. Traces are often logged in a tabular format and follow an underlying structure such as a graph, where the graph can be arbitrarily deep and wide. This means that it is non-trivial to represent valid traces as

| | EDGE ID | Src. | Dest. | Type | Start (ms) | Finish (ms) |
|---|---|---|---|---|---|---|
| ❶ | 0 | Client | Front end | HTTP | 0 | 24 |
| ❷ | 0.1 | Front end | Authentication | RPC | 0 | 1 |
| ❸ | 0.2 | Front end | Feed | RPC | 1 | 23 |
| ❹ | 0.2.1 | Feed | Posts | DB | 6 | 15 |

Figure 1: A simple social network application consists of eight microservices (Huye et al., 2023). Each user request triggers a sequence of microservice calls, forming a microservice call graph. The red lines represent the microservice call graph for a user request. Microservice call graphs are commonly logged in a tabular format, as shown in the figure on the right. Each row in the table represents a communication between two microservices, with the details of the communication logged as features in the columns.

text sequences, which is the format best suitable for modern autoregressive LLMs. Moreover, there are often complex implicit constraints in trace data that rely on relationships between multiple trace features. For example, an application process's start time must be earlier than the start time of all the child processes it spawns, while the parent process's end time must be later than the end time of its children; such constraints need to hold at all nodes in the application's graphical representation.

Our proposed approach adapts general-purpose LLMs to generate synthetic system traces, with a particular focus on *microservice call graphs*, a type of trace with rich directed acyclic graph (DAG) structures. We represent these graphs in a text-based format suitable for LLMs, enabling their use in trace generation. One of our key innovations is to generate call graphs with structural constraints by *recursively generating subgraphs*, or layers. This approach allows the model to break down the complex task of reasoning about hierarchical graph structures and complex constraints into multiple easier tasks. To further enhance the model's ability to follow structural constraints and to meet user-requested attributes, we perform instruction tuning. During this phase, the model learns to explicitly generate a series of *intermediate instructions* between recursive layer generation steps, performing simple arithmetic and logical checks to ensure adherence to the desired structure.

We demonstrate the effectiveness of our approach by fine-tuning Llama-2 7B (Touvron et al., 2023) with our method on microservice trace data and performing a series of evaluations. The results show that the proposed recursive generation and use of intermediate instructions significantly enhance the model's ability to generate valid outputs for complex (i.e., deep and wide) call graph structures. When compared to traces from learned generative models and a probabilistic model, synthetic traces produced by our model more closely match the distribution of real traces. We further show that our fine-tuned model performs well when used for downstream tasks for trace feature prediction compared to Llama-3.1 405B, one of the state-of-the-art LLMs.

We summarize our key contributions below:

- We introduce a novel method for creating valid and rich synthetic microservice traces using LLMs: (1) We *recursively generate* layers of subgraphs along with prompts for subsequent layers and (2) We also align the LLM to describe the generated layers with *intermediate instructions*.
- We show that recursive generation and intermediate instructions improve the validity of synthetic traces. Also, synthetic traces generated by our model are more realistic regarding distribution similarities than those from a baseline generative model and a handcrafted expert model.
- We demonstrate that synthetic traces can effectively substitute real data for training microservice management tasks. Furthermore, our model generalizes to unseen user-requested attributes during inference and can be further adapted for downstream tasks like infilling missing trace data.

## 2 BACKGROUND

**Microservice Call Graphs.** In modern software architecture, an application is typically constructed as a constellation of multiple microservices (Gan et al., 2019; Luo et al., 2022; Huye et al., 2023), each with specific functionalities and dependencies on one another. When users interact with these applications, for instance, by sending HTTP requests to web servers, a complex sequence of communications among these microservices is triggered. Thus, a user request induces a microservice *call graph*, which maps the control/data flow and dependencies among the microservices involved in fulfilling the user's request.

Figure 1 is an example of a social network application deployed with several microservices (8 in total). In the figure, the red arrows indicate communications between microservices involved in processing the user's request. The request is sent to a microservice (e.g., "Front end" in Figure 1) and waits for the communication to terminate. If the microservice requires additional communication to handle the request, then it triggers another microservice call (e.g., from "Front end" to "Authentication" in Figure 1). These communications triggered by a user's request form a microservice call graph with four microservices. The vertices of the graph correspond to microservices (or the client), while the edges correspond to API calls invoking the microservices. Note that some edges are not part of the call graph as the corresponding microservices are not invoked in processing this particular request.

Each call graph can be represented as a tabular log trace with a textual description of the features of each API call (i.e., edges), including the source and destination of the request, type of request (e.g., HTTP and RPC), and start/finish time. As call graphs have a *hierarchical structure*, the tabular trace should preserve the parent-child relationships by ensuring that the child's source matches the parent's destination. Moreover, the start and end times of each call should be consistent with each other: (1) the start time of a microservice must be earlier than its finish time, and (2) the parent-child relationships must be honored, i.e., the parent's start time must precede the child's, and parent's finish time must follow the child's. Finally, the IDs within a call graph (dot-decimal numbers provided for each call) must also be hierarchically connected to form a DAG structure.

**Synthetic Trace Generation using Machine Learning.** The analysis of microservice traces plays a pivotal role in improving the performance and reliability of services, and guides techniques that enable high-performance and efficient use of the underlying machines. Representative use cases include critical path analysis (Zhang et al., 2022), anomaly detection (Xie et al., 2023), root cause analysis (Ikram et al., 2022), cluster management (Qiu et al., 2020), and cluster scheduling (Singhvi et al., 2021). Unfortunately, access to traces remains challenging due to business and privacy concerns.

Given the importance and limited availability of public computer system traces, including microservice traces, several recent studies have explored generative models for synthetic trace generation. Existing works (Lin et al., 2020; Jiang et al., 2023) leverage GAN (Goodfellow et al., 2014) and diffusion (Ho et al., 2020) models to generate network packet traces, while other work (Bergsma et al., 2021) uses LSTMs (Sherstinsky, 2020) to generate virtual machine workload traces. Even though the generative models have shown effectiveness in each domain, the methods are used only for predicting specific fields or following training data distributions without conforming to structural constraints. These methods do not apply to microservice call graphs because they cannot handle the hierarchical structures of the call graphs.

Since traces have specific structures that can be represented in tabular form, machine learning methods for synthetic tabular data generation could be applied to synthetic trace generation. Recent approaches, such as TVAE (Xu et al., 2019) and GReaT (Borisov et al., 2023), leverage VAE (Kingma & Welling, 2013) and language models to advance synthetic tabular data generation techniques. However, these methods have limitations when applied to microservice traces, as they do not account for the hierarchical structure of call graphs within tabular representations. We provide a detailed comparison with tabular data generation methods in §4.

## 3 TRAINING LLMS TO GENERATE MICROSERVICE TRACES

Our goal is to train a generative model for microservice call graph traces. We want to allow end-users to simulate various scenarios, such as stress-testing a novel software application or feature, by conditioning the model's output on user-requested attributes including the application being invoked, the number of microservice communications (i.e., graph edges), and the overall latency of the application. Given the limitations of existing trace generation approaches, we turn to LLMs, which are transformer-based (Vaswani et al., 2017) models with billions of parameters. We initialize our model from a general-purpose LLM pre-trained on a large and diverse text dataset, as these models have shown effectiveness when adapted for specialized domains such as proteins (Shen et al., 2024) and code (Roziere et al., 2023). In addition, LLMs can be conditioned in a variety of arbitrary manners, including natural language prompting (Ouyang et al., 2022) and structured input sequences (Borisov et al., 2023).

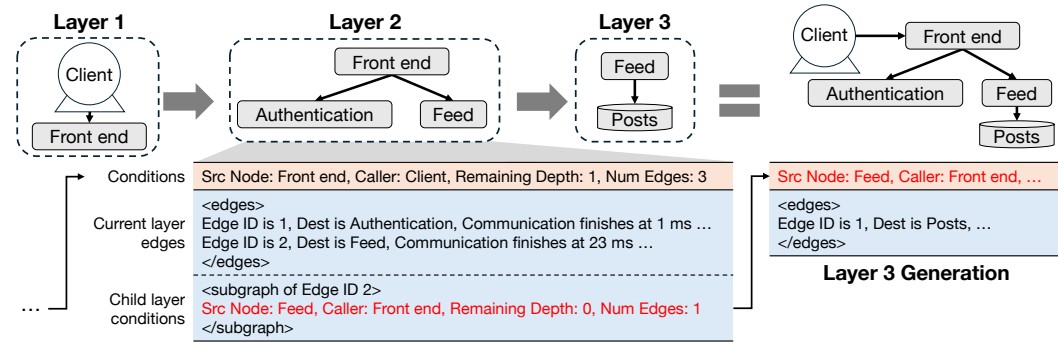

Figure 2: Overview of the recursive generation method with a simplified example. The model uses conditions generated in *Layer 1* (e.g., source node, caller, number of edges) to generate two edges in *Layer 2*, one leading to *Authentication* and the other to *Feed*. The model also generates starting conditions for the next layer, beginning from the *Feed* microservice. This recursion continues until all edges in *Layer 3* are generated.

This section presents our approach for training an LLM to generate microservice call graphs. We train our model in two stages. In the first stage, we pre-train the model to learn the complex interactions between the vertices and edges in real call graph data. We describe how we encode call graphs, stored as tabular data, into a text format that can be tokenized and processed by the LLM. Then, we detail a novel approach to improve the model's generation of large, complex graph structures. We propose to decompose the graph generation task into a series of simpler, recursive subgraph generation tasks that allow the model to reason about local features while respecting global structure. In the second training stage, we fine-tune the model to follow user instruction and allow flexible generation of call graphs with desired attributes. During instruction tuning, we propose to include a series of natural language reasoning steps that reinforce the model's ability to adhere to constraints during inference.

## 3.1 PRE-TRAINING

We pre-train our model on call graphs using an autoregressive language modeling objective. This stage adapts the general-purpose LLM, which was previously trained to model natural language text sequences, to the more specialized domain of microservice call graphs.

### 3.1.1 ENCODING CALL GRAPHS AS TEXT

LLMs expect sequences of text as input, so we must encode our dataset of call graphs into text-based representations before training our model. As detailed in §2 and shown in Figure 1, microservice call graphs are initially stored as tables. Rows represent edges (i.e., communications between microservices), while columns describe features for each edge. We follow the method proposed by GReaT (Borisov et al., 2023) and encode features in a natural language format, which requires minimal pre-processing and allows us to exploit the LLM's pre-training on diverse, natural texts. Our lossless encoding procedure preserves all necessary information to recover the unique graph that produced the tabular data. Besides edge features, we also encode global attributes of the call graph to serve as conditioning information for the model.

Tabular call graph $\mathbf{X}$ has $m$ columns of features with textual names $\{f_1, f_2, \ldots, f_m\}$ and $n$ rows of edges $\{\mathbf{x}_1, \mathbf{x}_2, \ldots, \mathbf{x}_n\}$. We denote the value of feature $j \in \{1, \ldots, m\}$ for edge $i \in \{1, \ldots, n\}$ as $v_{ij}$. We encode each edge $\mathbf{x}_i$ as a text sequence $\mathbf{t}_i = [t_{i1}, t_{i2}, \ldots, t_{im}]$, where $t_{ij}$ is a description of the $j$-th feature with the format $t_{ij} = [\phi(f_j), v_{ij}]$. Here, $\phi(f)$ encodes feature name $f$ into a text template with a subject-predicate structure to provide a natural language description of feature value $v_{ij}$. For instance, the encoding for edge 1 in Figure 1 would be [Edge ID is 0, Source is Client, Destination is Front end, Type is HTTP, Communication starts at 0 ms, Communication finishes at 24 ms]. We encode tabular call graph $\mathbf{X}$ to the equivalent text-based representation $\mathbf{t} = [\mathbf{t}_1, \mathbf{t}_2, \ldots, \mathbf{t}_n]$, formed as a sequence of the text-encoded edges $\mathbf{t}_i$. We note that the structure and constraints of the call graph only depend on the feature values and are invariant to the specific feature ordering imposed by the columns of the tabular

data. Therefore, during training, we randomly shuffle the order of the features within each edge as in Borisov et al. (2023) to remove any spurious associations that arise from position information.

Apart from individual edges, the overall call graph can also be described by attributes, including the maximum depth, the total number of edges, and the total communication latency. These attributes are useful for summarizing the complex interactions between edges and can be fed to the model as a prompt to condition call graph generation. Let call graph $\mathbf{X}$ have $r$ attributes with names $\{a_1, a_2, \ldots, a_r\}$ and corresponding values $\{w_1, w_2, \ldots, w_r\}$. We encode the attributes as a text sequence $\mathbf{c} = [c_1, c_2, \ldots, c_r]$, where $c_j$ is a description of the $j$-th attribute with the format $c_j = [a_j, \text{``} : \text{''}, w_j]$. See the *Conditions* shown in red in Figure 2 for a simplified example of text-encoded call graph attributes. We include the attributes at the start of each text-encoded call graph sequence and predict them along with the edge tokens during pre-training. Similar to the edge features, we randomly shuffle the order of graph attributes during training. We additionally drop each attribute independently with probability $p_{drop}$ to allow flexible prompting with arbitrary subsets of attributes.

### 3.1.2 Recursive Generation

We propose to break down the task of generating a call graph into a series of recursive *layer generation* tasks to handle complex structures. Starting from the initial attributes, or *prompt* $\mathbf{c}$, the task for the model at each layer is to generate the edges originating from the *Start Node* specified in the prompt. The model also generates a new prompt for the next layer based on the previous layer prompt and the edges generated in the current layer. This new prompt is then re-used to condition the model's output for the next layer. The recursive process continues until the requested attributes $\mathbf{c}$ are satisfied.

Formally, for an encoded call graph $\mathbf{t} = [\mathbf{t}_1, \mathbf{t}_2, \ldots, \mathbf{t}_n]$, we partition the edges $\mathbf{t}_i$ into a sequence of layers $[\mathbf{t}^1, \mathbf{t}^2, \ldots, \mathbf{t}^l]$, where $l \leq n$. Each layer is comprised of a sequence of edges that share the same parent (i.e., source) node, and no two edges are shared by layers. For call graph conditions $\mathbf{c}$ that describe $\mathbf{t}$, we introduce layer conditions $\mathbf{c}^j$, $j \in \{1, 2, \ldots, l + 1\}$. Layer condition $\mathbf{c}^j$ encodes the attributes of the remaining portion of the call graph after the sequence of layers $[\mathbf{t}^1, \mathbf{t}^2, \ldots, \mathbf{t}^{j-1}]$ has been generated, and we define $\mathbf{c}^1 := \mathbf{c}$ and $\mathbf{c}^{l+1} := \emptyset$. We decompose the conditional call graph distribution as a chain of conditional layer distributions:

$$p(\mathbf{t}|\mathbf{c}) = \prod_{k=1}^{l} p(\mathbf{c}^{k+1}, \mathbf{t}^k | \mathbf{c}^k) \tag{1}$$

In other words, the model predicts call graphs from user prompts iteratively layer-by-layer. For layer $k$ the model takes conditions $\mathbf{c}^k$ as input and produces the sequence of edges $\mathbf{t}^k$ followed by the conditions $\mathbf{c}^{k+1}$ of the next layer. The model-generated conditions $\mathbf{c}^{k+1}$ are then re-used as inputs to predict the next layer, $k + 1$. Figure 2 illustrates an example of a recursively generated call graph.

### 3.2 Instruction Tuning

We perform supervised fine-tuning after pre-training to improve the model's ability to generate call graphs following user instructions. Different from pre-training, we do not calculate loss for the model on the initial call graph attributes $\mathbf{c}$ (equivalent to the first layer conditions $\mathbf{c}^1$), which are now treated as a fixed prompt. The user can supply additional natural language instructions for the model, and in §4.4, we provide results for two types of additional instruction. We further supplement the instructions with additional prompts, which can be programmatically generated from a template based on the user-requested attributes, to aid the model's reasoning abilities, as detailed in §A.3. These prompts convert the attributes, including numbers, application ID strings, and other non-natural language inputs, into natural language instructions.

#### 3.2.1 Intermediate Instructions

We find that the model often struggles to generate consistent and correct next layer conditions $\mathbf{c}^{k+1}$ based on the current layer edges $\mathbf{t}^k$ and conditions $\mathbf{c}^k$ during recursive generation. For instance, the model may generate conditions that violate physical constraints, such as assigning a higher latency to a layer than the overall call graph. Inspired by recent work demonstrating that LLM ability improves when explicitly forced to reason step-by-step (Wei et al., 2022; Nye et al., 2021), we propose including a series of natural language reasoning steps that reinforce the model's ability to adhere to

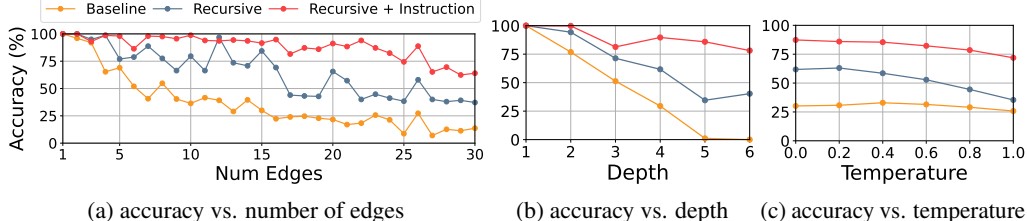

(a) accuracy vs. number of edges     (b) accuracy vs. depth    (c) accuracy vs. temperature

Figure 3: Call graph generation accuracy with varying (a) edges and (b) depth in prompt using greedy sampling. The plot in (c) shows the accuracy with varying sampling temperature. Accuracy measures the fraction of generated traces that are valid and follow the initial instructions. As shown, both recursive generation and instruction tuning help to increase the validity of the synthetic traces.

constraints. For example, we include a step-by-step calculation to (1) find the number of remaining edges based on the *Num Edges* attribute in $\mathbf{c}^k$ and the number of edges generated in $\mathbf{t}^k$, and (2) decide the *Remaining Depth* attribute in $\mathbf{c}^{k+1}$ from the same attribute in $\mathbf{c}^k$ (e.g., `Child's remaining depth = the current remaining depth – 1 = ...`). We include these *intermediate instructions* immediately before the next layer conditions $\mathbf{c}^{k+1}$ during instruction fine-tuning. We give an example of these reasoning steps in §A.3.

## 4 EVALUATION

We demonstrate the effectiveness of our method in two major aspects: (1) synthetic trace quality in terms of structural validity (§4.1), distribution similarity (§4.2), and usefulness to train and evaluate machine learning-driven microservice management tasks (§4.3), and (2) benefits from our use of LLMs in terms of instruction-following capabilities (§4.4) and downstream task performance (§4.5).

We initialize our model from Llama-2 7B (Touvron et al., 2023) and train with LoRA (Hu et al., 2022) on 1.36 million microservice call graph samples from the Alibaba v2022 dataset (Luo et al., 2022), corresponding to 1.1B tokens. We reserve 10% of these samples for validation. Instruction tuning datasets were created by randomly selecting 5% of the training graphs and reformatting them for instruction tuning. The training lasted four epochs, using a temperature of 0.8 and top-K of 50 for trace generation, unless otherwise specified. Further details on data preprocessing and training hyperparameters are provided in Appendix A. We compare synthetic trace quality with various structured data generation methods such as GReaT (Borisov et al., 2023) and TVAE (Xu et al., 2019), and downstream task performance with one of the state-of-the-art LLMs, Llama-3.1 405B.

### 4.1 STRUCTURED REASONING WITH RECURSIVE GENERATION AND INSTRUCTION TUNING

This experiment demonstrates how recursive generation and instruction tuning with intermediate instructions enhance LLMs' ability to accurately construct microservice call graphs. We evaluate our model by generating traces with specified `num_edges` and `depth`. A trace is deemed accurate if it correctly matches the specified `num_edges` and `depth` and adheres to all structural constraints, such as valid DAG formations and appropriate start/finish times for communications, detailed in Appendix B. We generate 50 samples for each (`num_edges`, `depth`) pair across ranges of $1 \leq$ `num_edges` $\leq 30$ and $1 \leq$ `depth` $\leq 6$.

**Baselines.** We compare our model (*recursive + instruction*) to Llama-2 7B models trained on text-encoded call graphs (1) without recursive generation and tuning with intermediate instructions (*baseline*) and (2) with recursive generation but no instruction tuning (*recursive*). Both baseline models are given `num_edges` and `depth` at the start of each sample during training (see Figure 9 for an example of a training sample for the *baseline* model). Baselines are trained using the same hyperparameters and number of tokens as our model. For the *baseline* model, we represent call graph traces as the tabular data format following the method in GReaT (Borisov et al., 2023).

**Results.** Figure 3a and Figure 3b present the accuracy of generated call graphs across varying numbers of edges and depths. Generally, as complexity increases (i.e., more edges or greater depth), the baseline model's accuracy decreases significantly—dropping below 25% for edges greater than 15 and nearing zero for depths above four. In contrast, the recursive generation model maintains higher

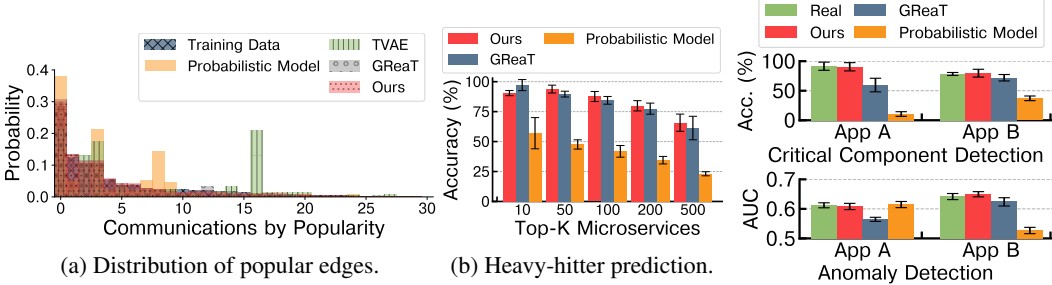

Figure 4: Distribution similarity between real and synthetic traces.

Figure 5: ML Model Performance (real vs. synthetic traces).

accuracies, approximately 30% and 35%, respectively. This improved performance is attributed to the model breaking down complex generation tasks into simpler, more manageable sub-tasks.

Figure 3c illustrates how accuracy varies with the temperature parameter during decoding. Both models show decreased performance as the temperature increases, but the recursive model consistently outperforms the baseline, maintaining about 10% higher accuracy even at a temperature of 1. Further, instruction tuning enhances model accuracy—from 23% to 36%— by directing the model to adhere to specific generation instructions, such as the number of edges per layer, which are outlined in §A.3.

## 4.2 SIMILARITY BETWEEN REAL AND SYNTHETIC TRACES

To evaluate the quality of synthetic traces, we compare similarities between real traces from the training dataset and synthetic ones. We generate 50K call graph traces using prompts generated by the validation dataset and compare to the call graphs in the validation dataset.

**Baselines.** We compare the following synthetic trace generation methods:

- **Llama-2 7B + tabular format (GReaT (Borisov et al., 2023))**: A Llama-2 7B model fine-tuned on the tabular data format of call graph traces (Same as *baseline* in §4.1).
- **Probabilistic model**: Probabilistic model based microservice call graph generators by Alibaba (Luo et al., 2021). The model is designed to follow the random distribution of different statistics, such as communication types and the number of children per depth.
- **TVAE (Xu et al., 2019)**: Tabular data generative model using VAE (Goodfellow et al., 2014). Since tabular data cannot be used to generate traces, we use the baseline only to compare distributions of popular edges. Also, to limit the training data size, we randomly choose 100k training samples from the trace dataset and use SDV (Patki et al., 2016) to train.

**Distribution of Popular Calls.** Realistic synthetic traces should mirror real-world communication patterns. To assess this, we analyze the distribution of calls, defined by the attributes *(Source, Destination, Communication type)*. Figure 4a illustrates the distributions of the 100 most popular calls generated by our method and the baselines, limited to the top 30 due to space constraints.

The KL divergence for traces generated by LLM-based approaches (ours and GReaT) is 0.16 and 0.11 respectively, indicating close similarity to the training data, whereas the probabilistic model's divergence is significantly higher at 3.84, due to its random selection processes. TVAE shows an intermediate divergence of 0.74, which is better than the probabilistic model but still less accurate than our method in capturing popular call distributions.

**Heavy-hitter Prediction.** The capability to generate heavy-hitter microservices—defined as top-$K$ microservices triggered in a sequence of call graphs—is critical for tasks such as resource optimization and anomaly detection in microservice management. In this experiment, we select 1K traces from the validation dataset and create instructions consisting of a service ID and call graph attributes such as depth and the number of edges. These instructions guide the synthetic trace generation for both the baseline and our models. We evaluate the accuracy by comparing the top-$K$ microservices between the synthetic and validation traces over 20 runs.

Figure 4b illustrates the accuracy for varying $K$ values, ranging from 10 to 500. Our method demonstrates robust performance, maintaining over 90% accuracy for $K \leq 50$ and 65% at $K$=500. The model trained with the GReaT method also shows robust performance, but slightly worse

performance with larger $K$ values. We believe that the performance gap results from the lack of capability to generate complex structures (§4.1), which can affect the distribution of generated traces. On the other hand, the probabilistic model starts at 59% accuracy for $K$=10 and declines to 23% at $K$=500, showcasing our method's capability to capture and replicate heavy-hitter dynamics in synthetic traces.

Additional evaluation results on the similarity of microservice branching (in-degree and out-degree) and response time distributions can be found in §D.3.

### 4.3 USEFULNESS OF SYNTHETIC DATA AS ML TRAINING DATA

The synthetic dataset is intended to serve as a substitute for real data in the training process. Thus, we assess how well state-of-the-art microservice management tasks for critical component extraction in FIRM (Qiu et al., 2020) and anomaly detection in TraceVAE (Xie et al., 2023), which use machine learning (ML) models, perform when the models are trained on the synthetic datasets. Specifically, the ML models are evaluated using real test data, and their results are compared to their original performance when trained on the real training dataset.

When choosing training data, we first select a subset of traces from real data and label them with corresponding conditions (e.g., critical microservices). To have similar distributions between synthetic and real traces, we extract instructions from the selected real traces and use them to generate synthetic traces. We do not include invalid call graphs using the same accuracy metrics in §4.1. We train each downstream task using 5K call graphs and evaluate it using 2K call graphs for testing. We consistently use the same test dataset derived from real data. We use the default hyperparameters set by FIRM and TraceVAE (Qiu et al., 2020; Xie et al., 2023), and set labels based on 99-percentile latencies. We run each experiment 5 times, varying the random seeds or test datasets, and report the average performace for the following two evaluation tasks. We use synthetic traces generated by GReaT and the Alibaba probabilistic model as baselines for comparing the performance of ML models.

**Critical Component Extraction.** For efficient resource management, FIRM (Qiu et al., 2020) predicts critical components (microservices likely to violate service level objectives (SLO)) from call graphs with support vector machines (SVMs) using latency-related features and determines additional resource types and amounts for the critical components. For our evaluation, we train SVMs to detect critical components using two popular applications (apps A and B) from our trace dataset. For each application, we randomly sample call graphs and train two SVMs: one with real data and one with synthetic data generated by our fine-tuned model. Figure 5 shows the accuracy of the models when evaluated on real call graphs from the test set. SVMs trained on synthetic data perform similarly to those trained on real data, differing by less than 1.5 percentage point. SVMs trained on synthetic traces from baselines exhibit an accuracy gap ranging from 6 to 81 percent points.

**Anomaly Detection.** For operators to efficiently diagnose system failures, anomaly detection models predict whether microservice call graphs include anomalous characteristics like irregular graph structure or time. We assess our synthetic data quality using TraceVAE (Xie et al., 2023), a variational autoencoder (VAE) model that detects anomalous microservices in terms of time consumption. We train TraceVAE models using real and synthetic trace data, similar to our previous experiment. Figure 5 reports ROC AUC for the models evaluated on real test data. Again, we see that training on synthetic data generated by our method consistently yields results comparable to those obtained from real data.

In conclusion, the synthetic traces from our method demonstrate similar performance to real traces and have the potential to be leveraged in various use cases, as demonstrated by the above evaluation tasks. We attribute our method's comparable performance with real data to its ability to generate complex structures and capture diverse characteristics. We also find similar results with two additional classification tasks by fine-tuning Llama-2 7B models. We describe the tasks and results in §D.4.

### 4.4 INSTRUCTION-FOLLOWING CAPABILITY

Enabling users to specify desired characteristics of synthetic data is crucial for trace generators. We assess our instruction-tuned model's capacity to reflect user-requested attributes in the generated traces accurately. We evaluate the model's ability to produce call graphs featuring specific attributes

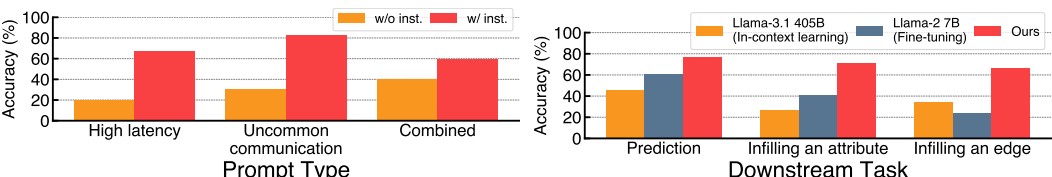

Figure 6: Instruction-following accuracy (%).      Figure 7: Downstream task accuracy (%).

(high latency and uncommon communications). Additionally, we explore the model's performance when prompted with a combination of these attributes not present in the training data.

When constructing the instruction tuning training datasets, we embed specific instructions to guide the generation of call graphs:

- **High Latency:** Instructions specify that call graphs should exhibit latencies above the 90th percentile (p90) of the training dataset's latency distribution, which varies by service. For example, the instruction might read: `Build a call graph with high latency`.
- **Uncommon Communications:** Instructions indicate that the call graph layer should include a communication occurring in less than 10% of the training data. An instruction example is: `Include an edge from (SRC) to (DEST) with (TYPE) communication type`.

We intentionally avoid combining these specific instructions in training samples to test the model's response to novel instruction combinations during inference.

**Results.** Figure 6 presents the instruction-following accuracy for high latency and uncommon communication. We assessed this by filtering 1K validation instructions to see how many generated call graphs met the defined criteria (e.g., exceeding p90 latency). We also compared these results against outputs generated without specific instructions to evaluate the impact of tailored prompts on model performance.

Additionally, we examine the model's performance when both instructions are combined in prompts, a scenario not covered in the training data. The model's ability to satisfy both conditions simultaneously, despite not being explicitly trained to do so, is detailed in the right of Figure 6. Higher accuracy in scenarios without specific instructions often results from inherent biases in attributes like service ID or the number of edges, which may align with the desired user outcomes.

### 4.5 ADAPTING MODELS TO SOLVE DOWNSTREAM TASKS

We extend our evaluation beyond generating synthetic traces, demonstrating the utility of our pre-trained model in performing downstream tasks related to microservice traces. The trace pre-trained model is adapted to each downstream task through additional fine-tuning. We focus on scenarios where partial information from distributed environment traces is available, emphasizing the challenges posed by incomplete data. This section compares our fine-tuned model with the standard Llama-2 7B, which lacks specific training on call graph data, and with Llama-3.1 405B by providing task descriptions and up to 16 examples in prompts (i.e., in-context learning (Brown et al., 2020)), to highlight the importance of domain-specific training.

**Predicting Uncommon Communication Patterns.** The task is to predict uncommon communication patterns (as in §4.4) based on the first 10 lines of a trace. We train the original Llama model and our adapted model for this binary classification task on 15K samples. Each sample's prompt comprised the first 10 edges of a real trace, with binary labels indicating the presence of uncommon communication patterns in the subsequent trace sections.

Results, detailed in the left of Figure 7, indicate that the original Llama-2 7B model achieves only 60.6% accuracy, suggesting insufficient training for recognizing uncommon patterns. Additionally, in-context learning with Llama-3.1 405B shows lower accuracy (45.6%), indicating larger models trained on typical internet data lack capability to solve tasks from domain-specific data. In contrast, our model achieves 76.8% accuracy, demonstrating its enhanced capability to interpret and predict based on partial trace data.

**Infilling Missing Data.** Missing data is common in large-scale trace logging, such as in Alibaba's microservice call graphs, where 67% of traces contain missing values (Huye et al., 2024). This

task focuses on fine-tuning our model to accurately infill missing data in microservice call graphs, considering partial information. Specifically, we conduct two separate experiments on infilling (1) a missing attribute and (2) a missing call connecting two layers.

In the first experiment, we construct a training dataset with 1.2K questions, each containing a sequence of edges with one attribute marked as [MISSING]. The missing value is the unknown ground truth for prediction, so these are multi-class classification problems. Attributes targeted include communication type (e.g., HTTP, RPC) or destination microservice. We evaluate the model on a 6K-sample test dataset, where our model demonstrated over 70% accuracy in predicting the correct attributes, significantly outperforming the accuracy of baselines by about 30% to 40% as reported in the middle of Figure 7.

The second experiment's dataset comprises 1K samples, each representing a pair of parent and child layers with a missing connecting edge tagged as [MISSING]. After training, we test both models on 5K test cases to generate the correct edge, ensuring the finish time matched or exceeded the start time. The right part of Figure 7 shows that while the original Llama-2 model scored only 24% accuracy and Llama-3.1 405B reached 34%, our model maintained a high accuracy of 66%, underscoring its robustness in more complex tasks.

These experiments demonstrate the capabilities of our trace pre-trained model to effectively adapt to handle infilling tasks that even large foundation models like Llama-3.1 405B cannot achieve.

## 5 OTHER RELATED WORK

**Adapting LLMs for Specific Domains.** Pre-trained LLMs are increasingly adapted for specialized domains due to their vast, diverse training datasets, which enable broad generalization capabilities. Examples include fine-tuning LLMs for programming (Roziere et al., 2023), quantitative reasoning (Lewkowycz et al., 2022), and semiconductor manufacturing (Liu et al., 2023). Our work is the first to apply this approach to computer system traces involving data with specific structures and constraints. Our focus is on generating synthetic trace data by fine-tuning these models to handle the specific requirements of this domain.

**Making Language Models Follow Instructions.** Recent advancements have focused on enhancing LLMs' ability to follow instructions through prompting (Li & Liang, 2021; Shin et al., 2020; Wei et al., 2022) and instruction tuning (Ouyang et al., 2022; Wei et al., 2021; Chung et al., 2022). These two sets of methods are relevant to our setting since they augment powerful pre-trained LLMs to improve their performance on new tasks. Our approach seeks to refine output expressiveness within set prompts, aiming for greater fidelity in synthetic data production.

**Multi-step Reasoning with LLMs.** Iterating with LLMs over multiple steps is an effective strategy to solve complex problems. For instance, Tree-of-thoughts (Yao et al., 2024) solves problems by decomposing into smaller thoughts and exploring diverse reasoning paths over different thoughts. Multi-step reasoning is also useful to handle long-context scenarios by summarizing iteratively (Wang et al., 2023) and diving into subproblems (Lee & Kim, 2023). In contrast to the above approaches, our approach learns to generate traces with specific structures and instructions for subsequent layers.

## 6 CONCLUSION

This paper presents a training method for pre-trained LLMs tailored for generating microservice trace graphs through a recursive call graph generation scheme complemented by instruction tuning with intermediate instructions. Our evaluation results demonstrate that our approach outperforms the baselines in generating accurate and valid call graphs and shows improved distributional similarity to real-world traces. In addition, we show that synthetic traces can effectively serve as a substitute for real data in training microservice management tasks, such as critical component detection and anomaly detection. They further highlight the effectiveness of instruction tuning in refining the generation of call graphs according to user-specified features and reveal the potential for using our model in various downstream tasks, such as prediction and data infilling, by further training the model. While this paper focuses primarily on microservice call graphs, our approach holds promise for broader applicability to other computer system traces with similar structural characteristics.

**Ethics Statement.** There are no ethical concerns raised by our work as the data used in this study is public and sensitive information has been anonymized.

**Reproducibility Statement.** We include the model and datasets used in experiments along with the steps to preprocess datasets. We describe training hyperparameters and data preprocessing steps in Appendix A. We include scripts to reproduce our experiments in the supplementary material.

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

# A  TRAINING DETAILS

## A.1  TRAINING SETUP

We train all models with $4\times$ A100 80GB GPUs in our cluster with the hyperparameters described in Table 1. We apply LoRA ((Hu et al., 2022)) adapters to query and key projection matrices of attention layers with $rank = 8$, $alpha = 16$, and $dropout = 0.1$. For the downstream task training in §4.5, we freeze the backbone model and only train the last classification layer for the prediction task. For the infilling downstream task, we use LoRA adapters with the same configuration as mentioned earlier.

## A.2  TRAINING DATA PREPROCESSING

From the Alibaba microservice v2022 traces (Luo et al., 2022), we use the first-hour call graph traces as our training data, which consist of 6434 unique microservices collected from more than 10 clusters. The traces are collections of API calls, where each API call includes communication information between the two microservices. Note that the dataset anonymizes the service and microservice names. Service ID is a nine-digit number starting with the prefix "S_" instead of using a real service name (e.g., social network), and microservice is a five-digit number starting with the prefix "MS_". We construct call graphs using the `trace ID` field (i.e., API calls with the same trace ID belong to one call graph). When constructing call graphs, we remove calls with missing information (e.g., destination microservice IDs are unknown) and remove call graphs that are not connected (e.g., missing edges). To remove redundancy, we filter out call graphs that have the same structure and fields (e.g., service ID, latency) for all API calls. The distributions of training data after removing redundancy are shown in Figure 8.

## A.3  TRAINING DATA EXAMPLES

From the call graph traces, we create text-based representations of call graphs as described in Section 3.1.1. First of all, Figure 9 is a training data example of converting a call graph into a tabular data format, which is the baseline in §4.1. At the beginning, we include high-level information about the call graph including service ID, the number of edges, and depth of the call graph. Each line inside the `<edges>` block corresponds to a call in a call graph. 6 fields exist for each call including the edge ID, source/destination microservices, communication type, and communication start/finish time.

Figure 10 shows an example training data sample for recursive generation as described in Section 3.1.2. Each sample consists of a sequence of layers, where each layer includes the edges and the conditions for the next layers. At the beginning of each layer, we provide high-level information to explain connections with the previous layers (e.g., `start_node`, `caller`), structure in the call graph (e.g., `remaining_depth`, `num_edges`, `start_edge_id`), and time-related information (e.g., `latency`, `start_communication_at`). Note that the number of fields in each edge is reduced from 6 to 5 since the edges share the same start node. Also, the edge ID field is an integer, not a dot-decimal number. For each next layer, the condition is described in each `<subgraph>` block starting with the edge ID to be extended.

Figure 11 is an example of instruction-tuning data. The instruction starts with a system prompt followed by conditions as in Figure 10. We further explain the condition in natural language formats along with user-requested features as studied in §4.4. In the output section, we include Chain-of-Thought scratchpads at the end of `<edges>` block and at the beginning of `<subgraph>` blocks, which elaborate on the number of edges to generate and constraints of subgraph conditions. For instance, the scratchpad includes descriptions regarding the depth requirement to let LLMs understand better that the depth field should be decreased by 1 from the current layer's depth.

As described in Section 3.1.1, we drop each call graph attribute randomly with probability $p_{drop}$. We set $p_{drop}$ to 0.9 for all attributes except for the service ID field, which is always kept ($p_{drop} = 1$), to ensure minimal conditioning.

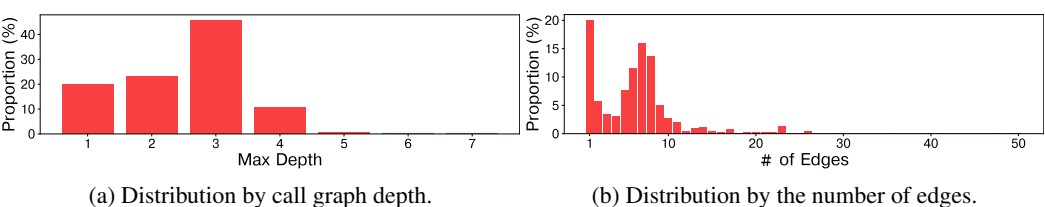

(a) Distribution by call graph depth.     (b) Distribution by the number of edges.

Figure 8: Training data distribution after preprocessing steps.

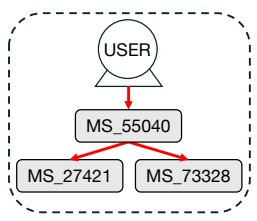

[GENERATE GRAPH]
num_edges:3/id:S_058367691/max_depth:2
<edges>
(source is USER,communication finishes at 2 milliseconds,communication starts at 0 milliseconds,type is rpc,destination is MS_55040,edge_id is 0)
(communication starts at 1 milliseconds, source is MS_55040,communication finishes at 1 milliseconds,destination is MS_27421,edge_id is 1,type is db)
(communication finishes at 2 milliseconds,type is db,communication starts at 1 milliseconds,destination is MS_73328, source is MS_55040 ,edge_id is 2)
</edges>

Figure 9: A training data sample of a call graph with 3 edges represented in tabular format.

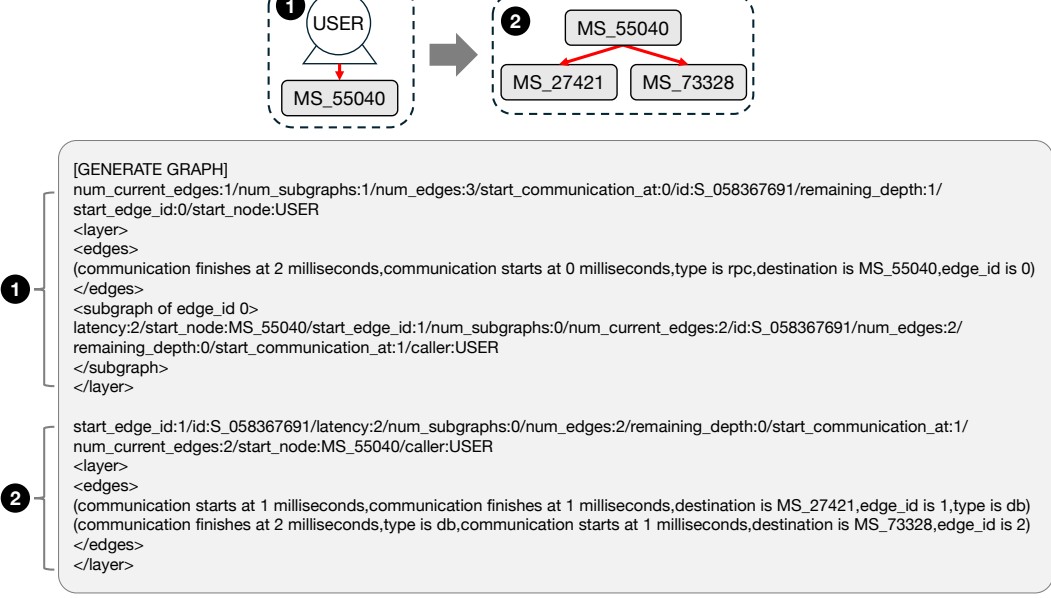

[GENERATE GRAPH]
num_current_edges:1/num_subgraphs:1/num_edges:3/start_communication_at:0/id:S_058367691/remaining_depth:1/
start_edge_id:0/start_node:USER
<layer>
<edges>
(communication finishes at 2 milliseconds,communication starts at 0 milliseconds,type is rpc,destination is MS_55040,edge_id is 0)
</edges>
<subgraph of edge_id 0>
latency:2/start_node:MS_55040/start_edge_id:1/num_subgraphs:0/num_current_edges:2/id:S_058367691/num_edges:2/
remaining_depth:0/start_communication_at:1/caller:USER
</subgraph>
</layer>

start_edge_id:1/id:S_058367691/latency:2/num_subgraphs:0/num_edges:2/remaining_depth:0/start_communication_at:1/
num_current_edges:2/start_node:MS_55040/caller:USER
<layer>
<edges>
(communication starts at 1 milliseconds,communication finishes at 1 milliseconds,destination is MS_27421,edge_id is 1,type is db)
(communication finishes at 2 milliseconds,type is db,communication starts at 1 milliseconds,destination is MS_73328,edge_id is 2)
</edges>
</layer>

Figure 10: A training data sample of a call graph with 3 edges for recursive generation.

**Instruction**

You are a trace generator that creates traces based on given requirements.
Requirements:
start_communication_at:0/start_node:USER/remaining_depth:2/num_current_edges:1/num_edges:4/latency:12/id:S_032647104
Conditions:
In each edge, communication start time should NOT be greater than latency 12 milliseconds
Generate subgraph instructions if necessary
the first start_communication_at should be requirement's start_communication_at 0
Also, communication should finish before latency 12 milliseconds
copy caller from requirement's start_node:USER
generate 1 edges following num_current_edges

**Output**

```
<layer>
<edges>
(edge_id is 0,type is http,communication starts at 0 milliseconds,destination is MS_57649,communication finishes at 12 milliseconds)
</edges>
num generated edges = the last edge id - the first edge id + 1 = 0 - 0 + 1 = 1
1 edges generated out of num_edges:4
num_remaining_edges = num_edges:4 - generated:1 = 3

generate subgraphs of edge:0
Subgraph constraints:
remaining_depth should be the requirement's remaining_depth:2 - 1 = 1
num_edges <= num_remaining_edges:3
copy start_node from edge 0 destination: MS_57649
<subgraph of edge_id 0>
remaining_depth:1/start_edge_id:1/num_edges:3/id:S_032647104/latency:12/num_subgraphs:1/num_current_edges:2/
start_node:MS_57649/start_communication_at:1/caller:USER
</subgraph>
now, num_remaining_edges is 3 - 3 = 0
finish generation
</layer>
```

Figure 11: A training data sample of a call graph layer for instruction-tuning.

Table 1: Training setup and hyperparameters.

| Model | Hyperparameter | Value |
|---|---|---|
| Pre-Training & Instruction Tuning | Optimizer | AdamW (Loshchilov & Hutter, 2017) |
| | Learning rate | 3e-4 with cosine scheduler |
| | Batch size | 64 |
| | Gradient clipping | 1.0 |
| Downstream Task Fine-tuning | Optimizer | AdamW |
| | Learning rate | 1e-4 with cosine scheduler |
| | Batch size | 2 |
| | Gradient clipping | 1.0 |

## B    Constraints in Call Graph Layers

In this section, we describe constraints to be met for each generated call graph layer to be correct. First of all, the generation results are considered invalid if the output does not have the valid format with `<edges>` and `<subgraph>` tags.

**Edges.**    For each edge, we check the following conditions. First of all, each edge should include the 5 fields: edge ID, destination, communication type, and communication start/finish time. Secondly, we check whether the right number of edges are generated as described in the condition. Third, the communication start time should be equal to or greater than the communication start time described in the condition, and should not be greater than the communication finish time of the edge. Lastly, the communication finish time should be equal to or less than the latency field in the condition.

**Next Layer Conditions.**    For the next layer conditions, we first check whether the next layer conditions should be generated or not. If the remaining depth field in the instruction is 0 or the number of edges that need to be generated is 0, no `<subgraph>` blocks should be generated.

Then, we check the validity of each field in the next layer conditions. First of all, the edge ID inside the `<subgraph>` block should be found in the edges generated in the current layer. For the depth, the remaining depth field should be less than the remaining depth of the instruction. Additionally, at least one of the resulting subgraphs must have a depth that is reduced by one compared to the original graph. For the `start node` and `caller` fields, they should be copied from the destination from the parent edge and the start node from the instructions, respectively. Lastly, we check the latency and communication start time by comparing the values to those of the parent edge. The latency of a child layer should not be greater than the communication finish time of the parent edge. Also, the communication start time of a child layer should not be less than the communication start time of the parent edge.

After generating both edges and the next conditions, we check if the sum of the number of edges matches the number of edges in the instruction.

## C    Limitations

In this section, we discuss a few limitations of our work and potential approaches to overcome the limitations. The recursive method improves call graph generation accuracy compared to generating the entire trace at once, but a key drawback is that previously generated edges are discarded, as only the conditioning information from the prior layer is passed to the next layer generation steps. Although dropping previously generated results has little impact on the output in microservice call graph generation, where direct neighbors exert the most influence (Zhang et al., 2024), we believe incorporating past information, such as previous layers or a time series of call graph traces, could be beneficial.

Furthermore, our method uses manually constructed instruction templates, which may lead to suboptimal generation quality, as we are not using the full potential of language models pre-trained with trillions of tokens (Touvron et al., 2023). Following the methods of Liu et al. (2024); Gunasekar et al. (2023); Li et al. (2024), we believe that diversifying instructions using LLM-generated output is a potential method to improve the ability of LLMs to follow user intentions.

Lastly, while protecting sensitive data during synthetic trace generation is an important research challenge, this paper does not address privacy concerns. Our framework assumes that the data used to train a model such as ours is curated at the source to remove sensitive attributes/values. As part of our future work, we plan to investigate whether our model exposes sensitive information and implement privacy-preserving techniques (e.g., differential privacy) during fine-tuning LLMs (Yu et al., 2022).

## D    Additional Evaluation Results

### D.1    Structured Reasoning Results in Detail

This section provides a more detailed analysis of the results from §4.1, accuracy to generate call graphs adhering to all structural constraints while matching the specified attributes in prompts (i.e.,

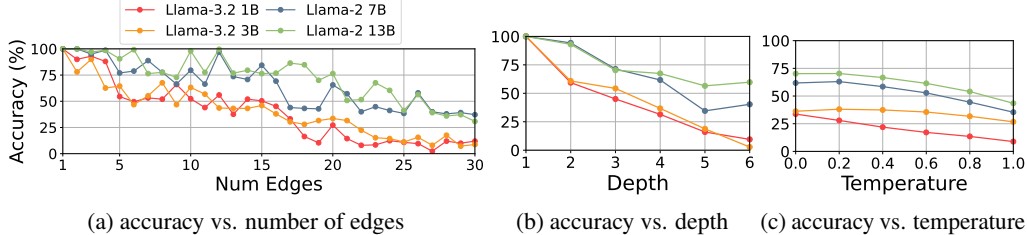

(a) accuracy vs. number of edges     (b) accuracy vs. depth    (c) accuracy vs. temperature

Figure 12: Call graph generation accuracy with varying model sizes. The plots in (a) and (b) show the accuracy varying edges and depth using greedy sampling, and (c) shows the accuracy varying sampling temperature.

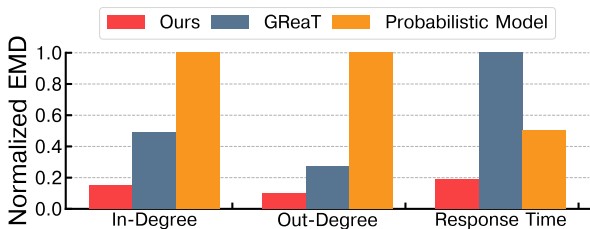

Figure 13: Distribution similarities in microservice branching (in-degree and out-degree) and response times between real and synthetic traces.

`num_edges` and `depth`). Figure 14 offers a closer look at Figure 3a and Figure 3b, where each grid point $(X, Y)$ represents accuracy for prompts with $X$ edges and a maximum depth of $Y$. Figure 14a, Figure 14b, and Figure 14c correspond to the same settings as *(baseline)*, *(recursive)*, and *(recursive + instruction)* from §4.1, respectively. The results in Figure 14 show that the recursive generation and instruction tuning improves accuracy across most combinations of (`# Edges`, `Depth`). However, some configurations in Figure 14b and Figure 14c exhibit lower accuracy, likely due to the distribution of training data in terms of edge count and depth.

In addition, we conduct an ablation study, where we remove *intermediate instructions* during instruction tuning to see the impact of *intermediate instructions* in generating correct call graphs. For instance, we remove equations and sentences that help to reason the properties to be generated (e.g., a sentence "`num generated edges = the last edge id – the first edge id + 1`" in Figure 11). Figure 15 reports the call graph generation accuracy varying the sampling temperature. Notably, removing the intermediate instructions during instruction tuning results in an approximate 13% decrease in accuracy across all temperatures, demonstrating the effectiveness of having intermediate reasoning steps during instruction tuning.

### D.2 STRUCTURED REASONING RESULTS VARYING MODEL SIZES

To evaluate the impact of model size on trace generation performance, we report the generation accuracy of models with varying numbers of parameters. Specifically, we compare four models: Llama-3.2 1B, Llama-3.2 3B, Llama-2 7B, and Llama-2 13B. Each model undergoes pre-training (§3.1) using the same training dataset (same as the *Recursive* setup described in §4.1).

Figure 12 presents the microservice call graph generation accuracy across different model sizes. Overall, models with a larger number of parameters demonstrate higher accuracy, with this trend being particularly evident in Figure 12c. Notably, models with more parameters perform better as the depth of prompts increases. For instance, the 13B model achieves a 20 percentage point improvement over the 7B model for inputs with a depth greater than 4 as shown in Figure 12b.

### D.3 MORE EXPERIMENTS ON SIMILARITY BETWEEN REAL AND SYNTHETIC TRACES

To further evaluate the effectiveness of our method in capturing the complexity of microservice interactions, we analyze the distribution similarities of microservice branching and response times using 10K synthetic traces. For consistency, we include only correct call graphs in the evaluation,

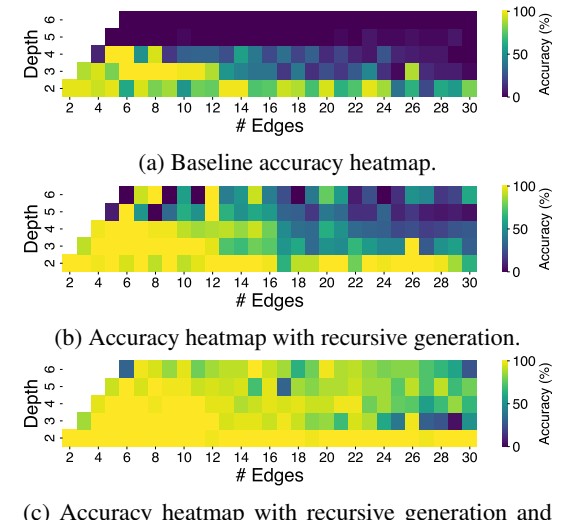

(a) Baseline accuracy heatmap.

(b) Accuracy heatmap with recursive generation.

(c) Accuracy heatmap with recursive generation and instruction-tuning.

Figure 14: Accuracy heatmap.

Figure 15: Accuracy to generate correct call graph structures with and without intermediate instructions during instruction tuning.

Table 2: Accuracy of prediction tasks by fine-tuning Llama-2 7B with real and synthetic traces.

| Accuracy (%) | High Latency | Uncommon Communications |
|---|---|---|
| Real | 68.3 % | 65.3 % |
| Synthetic | 67.1 % | 62.5 % |

following the accuracy criteria outlined in §4.1. The same baselines as in §4.2 are used, including GReaT and Alibaba probabilistic model. To extend the probabilistic model to include time-related fields, we augment it to generate response times by sampling from the training data statistics.

Figure 13 presents the distribution similarities for microservice branching and response times. We use normalized Earth Mover's Distance (EMD) as the similarity metric, ensuring comparability across fields with varying scales. *In-Degree* represents the distribution of the number of communications received by each microservice, while *Out-Degree* reflects the number of communications initiated by each microservice. *Response Time* measures the distribution similarity of the duration required to complete each communication. Across all three metrics, our method consistently achieves the closest results to the training data, achieving a 2.6x to 10x reduction in EMD compared to GReaT and the probabilistic model. We attribute its higher EMD values to an inability to generate complex call graph structures effectively.

### D.4 MORE EXPERIMENTS ON USING SYNTHETIC TRACES IN ML USE CASES

Building on the two evaluation tasks in §4.3, we conducted similar experiments using two classification tasks, fine-tuning the original Llama-2 7B models. We predict high latency in call graphs, defined as latency equal to or above the 90th percentile for each service, without providing latency-related information in the input data. Secondly, we predict uncommon communications in direct neighbors within a call graph, as defined in §4.4.

We fine-tune the original Llama-2 7B as a classifier by replacing the last layer with a classification layer and training only the last layer for one epoch. As in the experiments in §4.3, we train one model using real and one using synthetic data. Table 2 reports the test accuracy on real test data. Although synthetic traces have a slight accuracy drop compared to real traces, they still exhibit similar characteristics and can be effectively used in real-world tasks. For Llama-2 7B fine-tuning, We use a few thousand call graphs as training, validation, and test data (ratio 8:1:1) for each classification task and conduct a grid search over learning rates and batch size.

