# OpenReview forum: "Large Language Models as Realistic Microservice Trace Generators"
_ICLR.cc/2025/Conference — Submitted to ICLR 2025_

### Official Review · Reviewer_Yg3a · 2024-10-31

**Soundness:** 3
**Presentation:** 3
**Contribution:** 4
**Rating:** 8
**Confidence:** 4

**Summary:**

This paper proposes a method for generating synthetic microservice workload traces by fine-tuning large language models (LLMs) to replicate complex, hierarchical microservice call graphs. Using a recursive generation approach, the model creates layers of the graph step-by-step, preserving structural constraints to ensure realistic trace outputs. Additional instruction tuning enhances the model's ability to follow specific trace requirements and generate uncommon scenarios, making the synthetic traces suitable as substitutes for real-world data in downstream tasks like anomaly detection. The method is shown to outperform existing generative approaches, offering a promising solution for environments where real trace data is limited or privacy-sensitive.

**Strengths:**

Innovative Use of LLMs for Trace Generation: The proposed use of large language models (LLMs) for generating realistic microservice workload traces, specifically microservice call graphs, is promising, leveraging recent advancements in language models to solve challenges related to data availability and privacy in system performance analysis.

Recursive Generation of Complex Graph Structures: The proposed method addresses the hierarchical and recursive nature of call graphs, which many synthetic data generation techniques struggle to capture. By breaking down call graph generation into recursive layers, the model can better manage complexity and maintain structural constraints, which is essential for realistic trace generation.

Instruction Tuning for Enhanced Trace Validity: The authors incorporate instruction tuning, adding intermediate reasoning steps that enforce trace constraints. This strengthens the model's ability to follow user specifications and generate traces that respect the structural dependencies within microservice architectures, such as start and finish times in hierarchical call graphs.

Demonstrated Practical Application: The paper shows that synthetically generated traces can effectively replace real traces in downstream tasks, including microservice management tasks like critical component extraction and anomaly detection. This is a valuable contribution, as it shows that synthetic data can potentially reduce reliance on sensitive real-world data while still maintaining performance.

Comprehensive Evaluation and Benchmarks: The authors conduct thorough experiments that compare the recursive and instruction-tuned model against baselines, such as probabilistic models and other generative methods like GANs and VAEs. They provide performance metrics across various tasks (e.g., trace generation, infilling, and downstream prediction tasks), giving a clear view of the model's strengths and limitations.

**Weaknesses:**

Limited Exploration of Long-Term Dependencies: While the recursive approach is effective in handling hierarchical structures, it does not retain previously generated layers or edges in memory. This may limit the model's ability to capture long-term dependencies or sequential behaviors across more complex, larger traces, which could be important in certain applications involving extensive trace histories.

Reliance on Manually Constructed Instruction Templates: The use of hand-crafted templates for instruction tuning may restrict the model’s adaptability and efficiency. Generating diverse instruction formats, possibly with automated methods, might improve the model’s generalization and reduce the manual effort required to customize instructions for different scenarios.

Performance Trade-Offs in High Complexity Scenarios: The model demonstrates a drop in accuracy with increasing complexity, particularly with greater numbers of edges and layers in the call graphs. Although the recursive approach and instruction tuning mitigate some of this complexity, the performance could suffer in very large or deeply nested call graphs, limiting its effectiveness for the most demanding or detailed traces.

Synthetic Data Quality Compared to Real Data: Despite the high accuracy and validity of synthetic traces for downstream tasks, the paper notes a slight performance drop when using synthetic traces compared to real data in certain tasks. This suggests that, while synthetic traces are a viable alternative, they may still not fully replicate the nuances of real-world data.

Lack of Real-World Deployment Evidence: While the model is tested in a simulated environment and evaluated on trace data from a known dataset (Alibaba v2022), there is limited evidence of its performance in a live production environment. Real-world deployment could reveal additional challenges or performance limitations, especially under more varied or unforeseen conditions.

Potential Privacy Risks with Instruction-Tuned Models: The instruction tuning approach, while effective, could raise privacy concerns if used with highly specific instructions or in contexts where sensitive information may inadvertently be reflected in the generation outputs.

**Questions:**

Given that the recursive approach discards previously generated layers, how does this affect the model's performance in scenarios where long-term dependencies across multiple layers are critical? Could the authors provide additional insights or results on the effects of layer depth and sequence length on trace validity?

The paper mentions the use of hand-crafted templates for instruction tuning. Have the authors considered experimenting with automatically generated or dynamically adapted instructions to better match a broader range of conditions? Could this approach improve model generalization?

Have the authors considered or conducted tests of the model in real-world environments? While the model performs well on simulated data, deploying it in live production systems might reveal additional insights about practical constraints, resource usage, and unexpected errors.

The results suggest a performance drop with increased trace complexity. Can the authors clarify the main factors contributing to this decline? Is it due to limited model capacity, recursive generation limitations, or instruction tuning?

Instruction tuning, especially with user-specific attributes, might raise concerns about inadvertently learning or generating sensitive patterns. Did the authors take any measures to ensure privacy during training? Are there risks of exposing specific information, especially if used in sensitive environments?

The paper focuses on microservice call graphs, but this method may be adaptable to other hierarchical data. Do the authors foresee limitations in applying this approach to other domains, such as healthcare workflows or complex IoT interactions?

---

> ### Author Response · Authors · 2024-11-21
>
> Thank you for your comprehensive and thoughtful review. We truly value the time and effort you invested in evaluating our work. Below, we provide a point-by-point response to your concerns.
>
> > Given that the recursive approach discards previously generated layers, how does this affect the model's performance in scenarios where long-term dependencies across multiple layers are critical? Could the authors provide additional insights or results on the effects of layer depth and sequence length on trace validity?
>
> We acknowledge that the context of earlier layers is lost when generating the next layers except for conditions inherited from the previous layer. However, *since microservices in direct neighborhoods influence each other the most*, we did not observe the quality degradation of synthetic traces according to our evaluations. Furthermore, statistical analysis of training data reveals that the percentage of microservice calls that depend on grandparent calls to construct flows existent in training data is 0.32%. For the other cases, passing the previous layer’s information is enough to generate correct call graphs.
>
> > The paper mentions the use of hand-crafted templates for instruction tuning. Have the authors considered experimenting with automatically generated or dynamically adapted instructions to better match a broader range of conditions? Could this approach improve model generalization?
>
> We have yet to explore the use of automatically generated or dynamically adapted instructions, but we recognize their potential to enhance model generalization, as shown in other works [1, 2]. In future work, we plan to investigate this approach by leveraging recent advancements in large language models (LLMs), which could allow for the generation of more flexible and effective prompts for instruction tuning.
>
> [1] Textbooks are all you need (arxiv 2023)
>
> [2] Visual instruction tuning (Neurips 2024)
>
> > Have the authors considered or conducted tests of the model in real-world environments? While the model performs well on simulated data, deploying it in live production systems might reveal additional insights about practical constraints, resource usage, and unexpected errors.
>
> We have yet to deploy the model in live production environments, but we recognize the potential benefits of doing so. Deploying the model in production would allow us to leverage the history of previous executions encoded within the model to synthesize potential future scenarios. This could provide valuable insights not only into practical constraints, resource usage, and unexpected errors but also into potential adversarial workloads that may arise in real-world systems.

---

> ### Author Response · Authors · 2024-11-21
>
> > The results suggest a performance drop with increased trace complexity. Can the authors clarify the main factors contributing to this decline? Is it due to limited model capacity, recursive generation limitations, or instruction tuning?
>
> To evaluate performance with increased trace complexity, we conducted a new analysis on generation accuracy as the model scales. Experiments using Llama-3.2 1B, Llama-3.2 3B, Llama-2 7B, and Llama-2 13B (Appendix D.2) revealed that larger models generally achieve better performance. Specifically, models with more parameters demonstrate higher generation accuracy, particularly in handling complex inputs with greater depth. For example, the 13B model demonstrates a 20 percentage point advantage over the 7B model when handling inputs with a depth exceeding 4. Detailed results of these experiments can be found in Appendix D.2 of the paper.
>
> To further explore language models' ability to learn complex traces, we plan to apply reinforcement learning-based approaches (e.g., RLHF) as one of our future steps, incorporating failed generation cases as penalties or negative examples.
>
> > Instruction tuning, especially with user-specific attributes, might raise concerns about inadvertently learning or generating sensitive patterns. Did the authors take any measures to ensure privacy during training? Are there risks of exposing specific information, especially if used in sensitive environments?
>
> We acknowledge that safeguarding sensitive data during synthetic trace generation is a significant research challenge. Synthetic traces that maintain key real-world characteristics while protecting privacy can encourage greater transparency from companies, strengthen further research into microservice management and create stronger industry-academia collaboration. As addressing privacy concerns is not the primary focus of this paper, we intend to evaluate whether our model unintentionally reveals sensitive information in future work. In addition, we plan to incorporate privacy-preserving methods, such as differential privacy, during the fine-tuning of LLMs [3].
>
> [3] Differentially Private Fine-tuning of Language Models (ICLR 2022)
>
>
> > The paper focuses on microservice call graphs, but this method may be adaptable to other hierarchical data. Do the authors foresee limitations in applying this approach to other domains, such as healthcare workflows or complex IoT interactions?
>
> Our proposed method could be adapted for different types of computer system traces besides microservice call graphs. Especially, system traces with hierarchical relationships, such as application function call traces, are our potential targets to extend our method. We believe that our method can be applied to other hierarchical data from healthcare and IoT domains.
>
> The primary requirement for adapting our LLM-based method to different kinds of hierarchical traces is that we can encode the trace data as text without losing any information. Therefore, the bulk of the modifications would concern the specific text-encoding methods we would need to devise to represent the traces. We would likely be able to keep the subject-predicate structure used to represent instance features since these are agnostic to the type of underlying data structure. However, if the new trace modality does not follow a hierarchical structure, we would replace the intermediate instruction and recursive generation with more appropriate schemes as needed.
>
> In addition, if keeping the earlier contexts (e.g., parent information in a hierarchical structure) is important in other system traces, it would be necessary to pass previous generation results when using our recursive generation methods. As stated in our limitation section, providing past information through existing techniques, such as memory augmentation [4], would be essential to generalize our method to other system traces.
>
> [4] Recurrent Memory Transformer (NeurIPS 2022)

---

> ### Author Response · Authors · 2024-11-29
>
> Regarding privacy issues, we have provided a detailed discussion in Appendix C.
> Please let us know if you have any additional comments or questions before the discussion period concludes.

---

### Official Review · Reviewer_EVRN · 2024-11-03

**Soundness:** 3
**Presentation:** 3
**Contribution:** 2
**Rating:** 6
**Confidence:** 3

**Summary:**

The paper proposes to use LLMs for the generation of synthetic micro-services call-graph traces with the aim of providing richer datasets for downstream tasks in the microservice management domain while overcoming the privacy concerns of service providers. To enhance the quality of generated call graphs, authors rely on two key ideas. First, they generate call graphs using a tabular representation, i.e., a table with each row being an edge with given features such as source, destination, type, start time, end time, encoded in a subject-predicate structure similar to the concept used by the GreaT tabular data generator and include a series of preconditions. Second, they generate the call graph in a recursive top-down manner with intermediate instructions, which helps the model to generate edges satisfying consistency constraints, e.g., end time not later than the overall call graph end time. Using these, they fine-tune a Llama2 8b model and evaluate the synthesized call graphs both in terms of statistical similarities with respect to the training data and of utility in downstream tasks, i.e., when used to train other machine learning models also compared to vanilla LLM and prior art trace generators.

**Strengths:**

+ The data synthesized by the proposed generator based on fine-tuned LLM with the use of recursive call graph generation is able to match the performance of real data traces when used to train ML models.
+ The LLM is able to incorporate specifications from users.
+ Ablation study to evaluate the impact of recursive generation and intermediate instructions

**Weaknesses:**

- One of the motivations for using synthetic data is to preserve privacy, but how much LLMs can or can not leak private information is still an open research question.
- Since the paper aims to generate (call) graphs, it would have been nice to include as a baseline a generator for attributed graphs.
- While authors use three baselines (TVAE and GreaT and a probabilistic model) when studying and comparing the distribution similarity between real and synthetic traces, the results on downstream utility are only performed against real traces. Moreover, here, the results are not very different from the ones obtained by GreaT.

Minor:
Figure 4a is a bit hard to read; I suggest plotting the data differently.
It would be nice to include confidence intervals in the plots of Figure 3.

**Questions:**

Please consider adding a baseline from the field of graph generators such as DiGress or similar and extending the result from Section 4.3 to the (at least a couple of the better performing) baselines.

**Details Of Ethics Concerns:**

Please consider adding a baseline from attributed graph generators such as DiGress and extending the Section 4.3 results to cover the baselines.

---

> ### Author Response · Authors · 2024-11-21
>
> We appreciate the reviewer’s thoughtful feedback. Below, we address your concerns and comments. Similar comments have been consolidated into a single section for clarity. We also intend to address the reviewer’s comments on the figures (Figure 3, 4a) in the final version of the paper.
>
> > Since the paper aims to generate (call) graphs, it would have been nice to include as a baseline a generator for attributed graphs.
>
> > Please consider adding a baseline from the field of graph generators such as DiGress or similar and extending the result from Section 4.3 to the (at least a couple of the better performing) baselines.
>
> > While authors use three baselines (TVAE and GreaT and a probabilistic model) when studying and comparing the distribution similarity between real and synthetic traces, the results on downstream utility are only performed against real traces.
>
> While microservice call graphs can also represented as graph representations, we did not choose graph generators such as DiGress as our baselines for the following reasons:
>
> * Graph dataset scale: Our dataset includes nodes with more than 6,000 microservices, but existing graph generation models such as DiGress struggle to scale to large graphs [1].
>
> * Parallel edges: Microservice call graphs include multiple edges with the same source and destination nodes, while existing generative graph models assume a maximum single edge between two nodes.
>
> Instead, we included additional baselines in Section 4.3 using GReaT and the Alibaba probabilistic models. As illustrated in Figure 5 of the revised paper, training ML models on synthetic data generated by our method consistently delivers performance comparable to that achieved with real data (showing less than 1.5 percent points gap for all cases), while the baselines show lower accuracy or inconsistent results (showing up to 81 percent points gap). These results highlight our method's ability to generate synthetic traces that closely resemble real ones.
>
> [1] Sparse Training of Discrete Diffusion Models for Graph Generation (arxiv 2024)
>
>
> > Moreover, here, the results are not very different from the ones obtained by GreaT.
>
> Existing results on heavy hitter prediction from GReaT show a gap of up to 5 percent points compared to our method, while yielding comparable results for popular call distributions. To further demonstrate the effectiveness of our method compared to the GReaT baseline, we added new evaluations in Appendix D.3, focusing on distribution similarities in response time and trace branching (in-degree and out-degree). In summary, synthetic traces generated by our method exhibit *much greater similarity, achieving up to 5.3x reduction in Earth Mover’s Distance (EMD) compared to GReaT*.
>
> Additionally, as explained in the previous response, we incorporated baselines in Section 4.3 using GReaT to highlight the effectiveness of our method in generating synthetic data that closely resemble real traces.
>
> > One of the motivations for using synthetic data is to preserve privacy, but how much LLMs can or can not leak private information is still an open research question.
>
> While protecting sensitive data during synthetic trace generation is an important research challenge, this paper does not address privacy concerns, as we select training data from a curated dataset where sensitive data is already encrypted and/or elided. Note that in general, our framework assumes that the data used to train a model such as ours is curated at the source to remove sensitive attributes/values.
>
> We plan to investigate whether our model exposes sensitive information as part of our future work. Also, we aim to implement privacy-preserving techniques, such as differential privacy, during the fine-tuning of LLMs [2].
>
> [2] Differentially Private Fine-tuning of Language Models (ICLR 2022)

---

> > ### Comment · Reviewer_EVRN · 2024-11-25
> >
> > Regarding privacy, if the dataset is already curated to remove any privacy-leaking information, one might ask why privacy is used as a motivation.
> > I appreciate, however, the effort to include more baselines and to motivate the lack of using graph generators such as DiGress. Since the size of the call graphs seems to play an important role, it would be nice to add some general statistics on this when introducing the dataset.
> >
> > I going to updated my score.

---

> > > ### Author Response · Authors · 2024-11-29
> > >
> > > Thank you for actively engaging in the discussion. We've included the privacy-related discussions in Appendix C for further reference. Please feel free to share any additional feedback.

---

### Official Review · Reviewer_KLHo · 2024-11-04

**Soundness:** 1
**Presentation:** 2
**Contribution:** 2
**Rating:** 3
**Confidence:** 3

**Summary:**

This paper introduces a method for generating synthetic microservice call graphs using large language models (LLMs). It attempts to address the difficulty of obtaining real-world traces by training LLMs to produce hierarchical and constraint-abiding call graphs recursively, breaking down the complex task into simpler sub-tasks. The authors also employ instruction tuning to align model outputs with specific trace features, in order to enhance the model's ability to generate valid and realistic traces. The paper evaluates the proposed approach by substituting for real-world data in system management tasks and adapt to downstream tasks like predicting trace features and infilling missing data.

**Strengths:**

The paper introduces an approach to generating synthetic microservice call graphs using large language models (LLMs), which is an interesting idea for system workload tracing. It attempts  to handle complex and arbitrary hierarchical structures and implicit constraints within microservice call graphs through a recursive generation method. The paper highlights the potential of synthetic traces to replace real-world data in optimizing and tuning system management tasks, offering significant advantages in terms of privacy and data availability. The method of this paper is of a certain level of innovation and practical value.

**Weaknesses:**

1. I have serious doubts about the motivation of this paper: Do we really need synthetic traces in the microservices trace domain?
    1）None of the three "synthetic trace generation" methods mentioned by the authors are for generating microservices traces: (Bergsma et al., 2021) is for generating cloud workloads, and (Jiang et al., 2023; Yin et al., 2022) are for producing network traces.

   2）The authors only made a brief statement about the motivation, "Obtaining real-world traces is often hindered by privacy concerns and their general unavailability," without providing any arguments or details. Moreover, the authors' method also requires "1.36 million microservice call graph samples" for training and validation, which is contradictory.

   3）In fact, in enterprises with microservices as the basic architecture, trace data is abundant and easily obtainable. There is no issue of insufficient data. If privacy prevents the use of this data, then the authors' method would also be inoperable.

   4）Furthermore, there are many microservices simulation systems, such as TrainTicket (X. Zhou, X. Peng et al., “Fault analysis and debugging of microservice systems: Industrial survey, benchmark system, and empirical study,” TSE’18, 2018). These systems also consume far less resources than the "4xA100" used in this paper.

2. The authors' experiments are insufficient and fail to prove the effectiveness.
    1) As a key metric for measuring the quality of generated traces, "Accuracy," the authors did not elaborate on its definition or how to count the "valid following the initial instructions." The authors also did not introduce the "initial instructions."

    2) We know that the calling relationships between microservices in real environments are very complex, so to verify the accuracy of the generated traces, many aspects need to be validated, not just "Distribution of Popular Calls" and "Heavy-hitter Prediction." For example: response delays of instances, branching of traces, etc.

   3) Figure 3 shows that only when the number of edges is less than 5 or the depth is less than 2 can the accuracy of the generated traces be guaranteed to be high. Otherwise, a certain proportion of inaccurate data will appear. I doult that if this inaccurate data is used as training data for anomaly detection or root cause analysis, it will severely affect the model's performance. For example, for anomaly detection, the authors only mentioned that TraceVAE performs similarly with real and synthetic data, suggesting that the authors provide more experimental details (such as whether the test sets are the same) and compare more algorithms.

   4) When comparing effects, it is not very meaningful for the authors to always compare with untrained LLMs. It is recommended to add comparisons with existing trace generation methods. For example, which has a greater overhead and better effect between the TrainTicket simulation system and this paper's generation method.

**Questions:**

1. The authors should further elaborate the necessity of synthetic trace generation in the microservices domain, especially when ample tracking data is typically available in microservices-based architectures? Additionally, if privacy concerns limit the use of real-world tracking data, why your synthetic trace generation method still need real-world data for training?

2. It is recommended that the authors compare the proposed method with existing trace generation methods or simulation systems, such as TrainTicket, particularly in terms of resource consumption and effectiveness, to demonstrate the advantages or unique features of their approach.

3. The authors are suggested to define the "accuracy" metric used in their experiments and explain how "valid following the initial instructions" is quantified. Furthermore, it is suggested to include additional validation metrics that capture the complexity of microservice interactions, such as instance response times and trace branching.

4.The authors are advised to provide more experimental details, including whether the test sets used for comparing real and synthetic data are the same, and whether the inaccurate data is include for training TraceVAE. Additionally, it is recommended to include comparative experiments with existing trace generation methods to prove the effectiveness of the proposed method.

5.Please add more experiments about, what are the implications that accuracy declines when the number of edges exceeds 5 or the depth exceeds 2? For example, given the presence of a certain proportion of inaccurate data in the generated traces, the authors should discuss how to manage these data, especially when using them as training data for anomaly detection or root cause analysis, to avoid affecting model performance.

---

> ### Author Response · Authors · 2024-11-21
>
> Thank you for your detailed and thoughtful review. We sincerely appreciate the time and effort you put into evaluating our work. We believe that your comments have helped improve the clarity of the paper’s contributions. Below, we respond to your questions point by point.
>
> > 1. The authors should further elaborate the necessity of synthetic trace generation in the microservices domain, especially when ample tracking data is typically available in microservices-based architectures? Additionally, if privacy concerns limit the use of real-world tracking data, why your synthetic trace generation method still need real-world data for training?
>
> We acknowledge that obtaining microservice traces is relatively easy with existing distributed tracing tools and microservice benchmark systems. However, acquiring traces from production environments, especially at large scale, remains a significant challenge. For example, the Alibaba microservice traces used in our evaluation were collected from more than 6,000 microservices across 10 datacenter-like clusters over 2 weeks – this represents a significant collection, curation, and analysis effort. Such at-scale trace collection is difficult, if not impossible, for medium-scale cloud tenants to conduct for their microservice applications due to the overhead and cost associated with tracing and analysis (leading to extremely low trace sampling rates, as little as 0.001% [1], and a high data loss ratio of up to 67% [2]); and it is clearly infeasible for researchers to replicate. In the absence of such quality trace data, companies (of various sizes) and researchers are hamstrung–e.g., they have to rely on outdated traces (such as the ~3-year-old Alibaba traces) to develop, test, and evaluate their microservice management solution. This is indeed the case with many studies and systems that have been developed over the past 2-3 years using the single trace [3, 4]. Unfortunately, such traces may not reflect the updated request and deployment patterns that a modern microservice application experiences.
>
> Simulating microservice call graphs is particularly valuable for generating other large-scale microservice benchmarks that can address research problems such as resource scheduling. This is the motivation behind work like [5], which builds synthetic call graph generators using probabilistic models. We extend this line of thought by showing the strong promise of leveraging language models as synthetic trace generators, given their ability to capture complex, diverse patterns in call graphs and enable detailed what-if analyses through natural language interfaces.
> Although a few companies and cloud providers release production traces, in many cases, trace releases are severely constrained or simply impossible due to proprietary concerns. Given this context, the question we seek is *how to produce diverse realistic synthetic traces, which are capable of incorporating important workload characteristics without disclosing sensitive information (e.g., user, application names), from the traces that a handful of companies (such as Alibaba) do make available*. This effort is naturally beneficial for the research community at large as well as for companies that don’t have their own trace collection infrastructure due to the aforementioned costs. *It is also beneficial for the company that sourced the original traces* (e.g., Alibaba) - this is because our generator can be used to produce interesting synthetic cases that can help test, debug, or develop new microservice management solutions for emerging workloads and applications.
>
> As a final remark, we note that the framework we designed would work as follows, which accounts for potential privacy concerns of companies where data is collected:
> * A company would release an instruction-tuned model following the recipe we advocate, as opposed to releasing raw or processed traces.
> * The data used to train the trace generator model would be curated at the source to avoid the inclusion of sensitive information.
>
> Of course, such models may still have remnant privacy risks, but we anticipate that to be of lesser concern than the privacy risks of raw data. In the future, we plan to conduct a more detailed study of whether models such as ours reveal sensitive data.
>
> [1] The Benefit of Hindsight: Tracing Edge-Cases in Distributed Systems (NSDI 2023)
>
> [2]  Systemizing and mitigating topological inconsistencies in alibaba’s microservice call-graph datasets (ICPE 2024)
>
> [3] Dissecting Overheads of Service Mesh Sidecars (SoCC 2023)
>
> [4] PERT-GNN: Latency Prediction for Microservice-based Cloud-Native Applications via Graph Neural Networks (KDD 2023)
>
> [5] Characterizing Microservice Dependency and Performance: Alibaba Trace Analysis (SoCC 2021)

---

> ### Author Response · Authors · 2024-11-21
>
> > 2. It is recommended that the authors compare the proposed method with existing trace generation methods or simulation systems, such as TrainTicket, particularly in terms of resource consumption and effectiveness, to demonstrate the advantages or unique features of their approach.
> Thank you for pointing this out. We provide our response below and plan to include it in the final version of our paper.
>
> Microservice benchmarks such as TrainTicket [6] are designed to analyze microservice architectures, their networking and system-level impact, and cluster management challenges. TrainTicket, in particular, emphasizes replicating fault scenarios identified through industrial surveys, and studying various debugging methodologies. Below, we outline a comparative analysis of TrainTicket and our proposed approach based on resource consumption and effectiveness.
>
> ### Resource Consumption:
>
>
> TrainTicket requires substantial resources to deploy its 41 microservices, and resource requirements grow with the complexity of the target deployment scenario. By contrast, our method primarily demands resources for training language models and generating synthetic traces. Once the training data is collected from deployed environments, no additional microservice deployment is needed, making our approach more resource-efficient over time for large-scale systems with numerous microservices.
>
> ### Effectiveness in Debugging:
>
> Through TrainTicket, developers can debug microservice applications by comparing successful and failed traces using visualization tools. However, this process requires developers to manually collect success and failure traces and apply debugging strategies—an approach that does not scale well with increasing microservice complexity. Furthermore, TrainTicket is limited in its ability to generalize across diverse microservice applications, necessitating additional effort for each unique application.
>
> Our method leverages language models to analyze microservice traces. These models, trained on complex datasets, enable anomaly detection and analysis through flexible interactions using natural language prompts. This adaptability allows for downstream tasks like prediction and attribute infilling without the need for manual trace collection and comparison, making it more scalable and versatile.
>
> ### Access to Application Logs:
>
> A notable strength of TrainTicket is its accessibility to application logs, which provide critical insights into microservice applications. While our current approach does not incorporate logs during training, we recognize their potential. Integrating logs into our training process could further enhance the capability of language models to simulate diverse and complex microservice scenarios.
>
> In summary, while TrainTicket excels at leveraging application logs and provides a robust framework for specific debugging tasks, our approach offers significant advantages in scalability, adaptability, and resource efficiency, particularly in environments with diverse and large-scale real-world microservice applications.
>
> [6] Fault Analysis and Debugging of Microservice Systems: Industrial Survey, Benchmark System, and Empirical Study (TSE 2018)

---

> ### Author Response · Authors · 2024-11-21
>
> > 3. The authors are suggested to define the "accuracy" metric used in their experiments and explain how "valid following the initial instructions" is quantified. Furthermore, it is suggested to include additional validation metrics that capture the complexity of microservice interactions, such as instance response times and trace branching.
>
> A trace is considered accurate if it precisely matches the specified `num_edges` and `depth` while satisfying all structural constraints, as detailed in the first paragraph of Section 4.1 and Appendix B. For instance, one structural constraint ensures that the communication start time for each edge does not exceed its finish time.
>
> In addition, we added new evaluations on distribution similarities in terms of response time and trace branching (in-degree and out-degree) in Appendix D.3. In summary, synthetic traces generated by our method demonstrate superior similarity based on Earth Mover’s Distance (EMD) metrics, achieving a 2.6x to 10x reduction in EMD compared to GReaT and the probabilistic model. This substantially improved performance highlights our method's effectiveness in capturing the intricate characteristics of microservice call graphs.
>
> > 4. The authors are advised to provide more experimental details, including whether the test sets used for comparing real and synthetic data are the same, and whether the inaccurate data is include for training TraceVAE. Additionally, it is recommended to include comparative experiments with existing trace generation methods to prove the effectiveness of the proposed method.
>
> To clarify, test sets used for comparing real and synthetic data are the same and the inaccurate data is not included for training TraceVAE. Relevant details have been added to the second paragraph of Section 4.3 in the revised paper.
>
> In addition, we added the results of GReaT and Alibaba’s probabilistic model as baselines to compare the effectiveness of applying synthetic traces for ML training in Section 4.3 of our revised submission. As shown in Figure 5, training ML models on synthetic data generated by our method consistently achieves performance comparable to using real data (showing less than 1.5 percent points gap for all cases), whereas the baselines exhibit lower accuracy or inconsistent results (showing up to 81 percent points gap). These findings demonstrate the capability of our method to produce synthetic traces closely resembling real ones.
>
> > 5. Please add more experiments about, what are the implications that accuracy declines when the number of edges exceeds 5 or the depth exceeds 2? For example, given the presence of a certain proportion of inaccurate data in the generated traces, the authors should discuss how to manage these data, especially when using them as training data for anomaly detection or root cause analysis, to avoid affecting model performance.
>
> While it is possible for inaccurate data to affect model performance, we ensure that no inaccurate data is used during training for downstream tasks such as anomaly detection by validating synthetic traces after their generation. Only valid traces are included in the training dataset for our experiments.
>
> On the other hand, removing inaccurate data from training datasets will influence the distribution of the training data and subsequently affect model performance. To reduce the amount of inaccurate data, we can apply a few strategies: 1) We can simply retry multiple times in the event of a failure. We observed that even a single retry improves the average accuracy from 71% to 84% when using a sampling temperature of 1.0. 2) We can further improve our model through other techniques, such as reinforcement learning by calculating penalties from the invalid generation results.
>
> To better understand the implications of accuracy declines, we conducted additional experiments comparing our method against other baselines, as described in Section 4.3 of our revised submission and the response to the previous question. Notably, the performance drops in ML models trained on GReaT’s synthetic traces imply how inaccurate data impacts ML model performance, with gaps ranging from 2 to 32 percent points compared to models trained on real data. In contrast, our model shows performance comparable to real data, with a gap of less than 1.5 percentage points, demonstrating the importance of accurate trace generation for ML model performance.

---

### Official Review · Reviewer_wddP · 2024-11-06

**Soundness:** 3
**Presentation:** 3
**Contribution:** 2
**Rating:** 6
**Confidence:** 2

**Summary:**

The paper proposes a novel approach to generate synthetic workload traces, specifically microservice call graphs using LLMs. The authors fine-tune LLMs to generate each layer recursively to make this generation a sequence of easy steps and finally apply instruction tuning to align the model with desired features. The authors evaluate their approach on Alibaba v2022 dataseta across many key dimensions and report significant gains.

**Strengths:**

1. The paper is well-written.
2. The authors solve an important problem of generating data for micros-service calls which might be difficult to acquire because of privacy and performance constraints.
3. This is an interesting domain of generating structured data.

**Weaknesses:**

1. The model evaluation is done for just one data, so the results might not be that generalizable to different datasets of varied characterisitics.
2. The model shows results for LLaMa-2-7B, but it does not evaluate for larger models. Will the proposed approach be as useful as the model scales.?
3. The model might overfit to certain structures if the training data does not capture diversity.

**Questions:**

1. Can you share the numbers of some other datasets>
2. Have you observed any pattern in the evaluation results as you scale the model?

---

> ### Author Response · Authors · 2024-11-21
>
> Many thanks for your thoughtful and valuable feedback. We address your concerns individually below and we have incorporated additional evaluations in Appendix D of our revised paper to address your feedback.
>
> > Have you observed any pattern in the evaluation results as you scale the model?
>
>
> Yes, we have observed clear patterns in the evaluation results as the model scales. To further illustrate this, we conducted additional experiments using Llama-3.2 1B, Llama-3.2 3B, Llama-2 7B and Llama-2 13B (Appendix D.2). The results of these experiments confirm that models with a larger number of parameters generally demonstrate improved performance.
> Specifically, larger models tend to produce higher generation accuracy, particularly in scenarios involving complex inputs with greater depth. For a detailed description of the experimental results, please refer to Appendix D.2 of the paper.
> > The model might overfit certain structures if the training data does not capture diversity.
>
> To ensure the training dataset includes diverse call graphs, we conduct preprocessing steps to remove redundant call graphs as described in Appendix A (briefly, we filter out call graphs with the same structures and field values). Furthermore, the training data includes call graphs from more than 6,000 microservices operating on 10 different clusters.
>
> Additionally, we have not observed evidence to suggest that our model is overfitted to certain structures in training data. As eval results comparing distribution similarity and evaluating the usefulness of synthetic data in ML model training show, synthetic traces generated with our model capture diverse and realistic characteristics in training data. We further evaluate our model’s capabilities, we report distribution similarities in terms of the trace branching and latency in Appendix D.3.
>
> > The model evaluation is done for just one data, so the results might not be that generalizable to different datasets of varied characteristics. Can you share the numbers of some other datasets?
>
> While we use one dataset by Alibaba traces as our training data for evaluation, the dataset includes diverse and complex traces collected from large-scale environments, as described in the answer to the previous question. We believe that the chosen dataset well represents traces with hierarchical structures and other datasets with similar structures can benefit from our trace generation method.
>
> Moreover, our proposed approach could potentially be adapted to various types of computer system traces beyond microservice call graphs. In particular, traces with hierarchical structures, such as application function call traces, represent promising targets for extending our method. To further evaluate the generalizability of our method, we are currently generating results with other trace datasets and plan to share an update soon.

---

> > ### Comment · Reviewer_wddP · 2024-11-23
> >
> > Thank you for the detailed answers to my questions. Most of my concerns are addressed. I have increased my score.

---

### Author Response · Authors · 2024-11-21

We are deeply grateful to the reviewers for their time and effort in helping us refine our paper. Your feedback and suggestions have been invaluable in improving the quality of our manuscript. We have carefully addressed each of your concerns and made substantial revisions accordingly. The blue text in the revised paper highlights the updates made during the discussion period. Below, we outline the key updates. Detailed responses to additional questions and concerns can be found in our replies to each reviewer.

### Baselines for Section 4.3: Synthetic traces as ML training data
In Section 4.3, we added baselines using GReaT and Alibaba probabilistic models. Figure 5 in the revised paper shows that ML models trained on our synthetic data achieve performance on par with real data (**showing less than 1.5 percent points gap for all cases**), unlike the baselines, which yield inconsistent results or lower accuracy (showing up to **81 percent points gap**). This demonstrates our method's high effectiveness in generating realistic synthetic traces.

### Distribution similarities in response time and trace branching
To further validate our method's ability to capture complex microservice characteristics, we added new evaluations in Appendix D.3, analyzing distribution similarities in response times and trace branching (in-degree and out-degree). The results, summarized using Earth Mover’s Distance (EMD) metrics, show that our synthetic traces closely resemble real ones, achieving a **2.6x to 10x reduction in EMD compared to GReaT and the probabilistic model**.

### Generation accuracy with model scaling
We conducted experiments with Llama models of varying sizes (1B, 3B, 7B, and 13B) to evaluate how graph generation accuracy improves with model scaling. The results confirm that larger models generally achieve better performance, particularly in handling complex inputs with greater depth. For instance, the 13B model achieves a 20 percentage point improvement over the 7B model for inputs with a depth greater than 4. Detailed results can be found in Appendix D.2.

---

### Meta-Review · Area_Chair_fan8 · 2024-12-19

**Metareview:**

The main advantage of the method described is that it would produce a synthetic dataset that does not directly expose the true operational data of any source. One problem with this goal is that there are no guarantees to avoid exposing private information from the training data, and their proposed solution to this issue is to further curate the data to remove private information. Of course, curating the seed data to that degree would also make it less useful as a representative set, and it’s not clear that the given evaluation setting actually reflects a realistic goal of removing private information in this way or what the tradeoffs would be. In particular, they do not directly evaluate the issues around privacy/memorization or model failures if the seed dataset used is not representative, which it must not be if it is curated for privacy reasons.

It was unclear why language models would produce more accurate distributions compared to probabilistic trace generators. My personal concern is that there may be significant memorization of the training data, which the authors do not check for in their evaluation.

Overall, I am somewhat ambivalent. It doesn’t seem any reviewer has close familiarity with the particular needs of customers for this product, and neither do I. Overall, this lack of clear familiarity with the requirements of such a product lead me to think that ICLR might not be an appropriate venue. There are no clear novel methodological contributions, so I have to evaluate this paper on the basis of clear practical contributions. I do not feel that the evaluation and analysis clearly demonstrate the advantage of this method in real-world settings, in particular because there is no explicit evaluation of whether the simulated distribution matches the diversity of the true distribution.

**Additional Comments On Reviewer Discussion:**

This was a difficult discussion to evaluate. All reviewers have low confidence scores except Yg3a, whose review is clearly edited---possibly de novo generated---by an LLM, and who did not reply to the authors' rebuttal. I am concerned by the amount of content in that review which is a simple restatement of methodological contributions and caveats explicitly given in the paper.

Reviewer KLHo, the only strongly negative voice, became somewhat rude during the discussion. However, I felt that several of their material objections held. The authors failed to directly respond to the weaknesses listed, instead only answering their questions. One of the questions the authors address is the request to compare to TrainTicket. The authors, rather than providing a direct comparison or evaluation, instead simply list some hypothetical advantages of their system over TrainTicket. I did not feel that this objection was adequately addressed.

The authors added multiple baselines comparing to probabilistic models during the discussion. Multiple reviewers raise their score in response. They also added new metrics for distribution similarity and accuracy of the generated data.

---

### Decision · Program_Chairs · 2025-01-22

Reject